# Intergenerational educational mobility in Bangladesh

**Mohammed Nazmul Huq**[1]*, **Moyazzem Hossain**[1], **Faruq Abdulla**[2], **Sabina Yeasmin**[3]

**1** Department of Statistics, Jahangirnagar University, Savar, Dhaka, Bangladesh, **2** Department of Statistics, Faculty of Sciences, Islamic University, Kushtia, Bangladesh, **3** Statistics Division, Bangladesh Bank, Dhaka, Bangladesh

* nhuq@juniv.edu

## Abstract

### Introduction

Social mobility is considered as an important indicator of the economic development of a country. However, it varies widely across geographical regions and social groups in developing countries like Bangladesh. This paper intends to evaluate the intergenerational mobility in Bangladesh across generations.

### Methods and materials

This paper considers a nationally representative sample survey of 8,403 respondents (rural: 5,436 and urban: 2,967). The male and female respondents aged 23 years and above were included in the sample. The education attainment of a son or daughter as compared to their father's education level was considered as the measure of intergenerational mobility. Transition probability matrix and different social mobility indices were used to find out the intergenerational education mobility in Bangladesh.

### Results

The findings reveal that approximately three-fourth (74.5%) of the respondents attained formal education, while more than half (58.3%) of the respondents' father was illiterate. The educational status of the respondents and their father who lived in urban areas was relatively better than who lived in rural areas. It is also observed that 91.2% and 81.6% of the intergenerational class movement was upward among sons and daughters respectively. The probability of a higher educated father will have a higher educated child is higher in urban areas than in rural areas of Bangladesh. The intergenerational mobility is higher in the primary, secondary, and higher secondary educational levels, though the illiterate and higher education levels are the least mobile classes. In addition, the limiting probabilities reveal that the chance of sending sons to schools by an illiterate father is less as compared to their daughters. Such difference is more obvious in the urban areas, i.e., it is highly likely that sons of the illiterate father are also illiterate.

**Data Availability Statement:** All relevant data are within the manuscript and its Supporting Information file.

**Funding:** This research received no specific grant from any funding agency in public, commercial or not-for-profit sectors.

**Competing interests:** The authors have no conflicts of interest to declare.

## Conclusion

Bangladesh has been progressing remarkably in recent years. To keep the pace of the ongoing economic development in the country, it is necessary to give more attention to the illiterate people especially the girls who live in rural areas. The authors anticipate that the findings will be helpful for the policymakers as the relationship between inequality and inter-generational mobility is vital for several aspects of the economic development of a country.

## Introduction

Mobility is an important concept in several branches of social science and economics [1]. Social mobility is deemed important in the process of economic development particularly in the underdeveloped and developing countries [2] as it provides information about equality of opportunity in society [3]. Social mobility is often measured using intergenerational mobility, which describes the association between parents and children in terms of different social strata including education, occupation, income [4, 5]. In the recent past, it has gained relevance due to increasing economic inequality [6] around the globe. The underlying principle to intergen-erational mobility is 'fairness', and this is achieved when every offspring has the chance of doing well, irrespective of their parents' social class. In a fair economy, as opportunities become more equal, mobility is likely to increase [7]. The greater economic mobility can also enhance social networks and stability, with people living in more mobile societies likely to be more optimistic about their future [8].

Intergenerational mobility has been the topic of research in developed countries for a long time. In the developed world, a comparatively high extent of social mobility across generations is correlated with the change from farm to non-farm field and expanded women's economic participation [9]. Educational extension, educational equalisation, and rapid structural reform in the economies of the US and Europe have led to greater social flexibility [10]. Moreover, there is a strong association between upward education and upward occupation mobility in England and Wales [11]. Recent evidence from the UK, as well as Anglo-German comparison, suggest that tertiary graduates from highly privileged parental social groups have a greater chance of joining top-level jobs as compared to their counterparts in less advantaged social groups [12]. Education policies may have an impact on intergenerational mobility [13]. How-ever, social mobility across developed countries varies based on their level of educational attainment, socio-economic growth and modernization [14]. For instance, total intergenera-tional mobility in Sweden increased during industrialization [15].

Mobility tends to be relatively lower in low-income countries compared to high-income countries. There is a wide variation in social mobility throughout regions and social groups in developing countries [16]. In low-income countries, parental socio-economic status has a sig-nificant effect on the child's future success [3, 17]. Intergenerational mobility is often linked with the parental background, particularly their income and education [18, 19], as wealthy parents spend, on an average, more resources for their children than poorer ones [20]. Gender also plays a crucial role as the mobility of girls is lower than that of boys in a developing coun-try like India [21].

Bangladeshi society can be sorted into different socio-economic strata. The recent macro-economic indicators suggest that people belonging to different social strata do not have equal opportunities to access economic activities. Although Bangladesh has been demonstrating commendable economic growth over the past years–the GDP per capita was $1,828 in 2018–

2019 as compared to $703 in 2008–2009 [22, 23], but job growth has remained slow. The employment elasticity was 0.55 between fiscal years 2005–06 and 2009–10 and dropped to 0.25 between fiscal years 2010–11 and 2017–18 [24]. Consequently, the high GDP growth over the past few years has been unable to lead to large-scale job creation.

In addition, the economic inequality measured by the Gini coefficient was 0.482 in 2016, up from 0.458 in 2010, representing high economic inequality in Bangladesh [25]. Such a high inequality skews opportunity [7] and limits the access of disadvantaged groups to productive economic activities. This vicious circle of high economic growth and high inequality generates evidence for weak social mobility status in Bangladesh. According to the Global Social Mobility Report 2020, Bangladesh has been ranked 78[th] among 82 countries in the world with a score 40.2 on a 100-point scale. Bangladesh is only ahead of Pakistan on the Global Social Mobility Index 2020 among South Asian countries. Sri Lanka ranked 59[th] on the index and topped the South Asian league chart. It is followed by India, ranked 76[th] globally [26]. With annual GDP growth of 8.15 percent in 2018–2019 [22], the low social mobility of Bangladesh clearly depicts that the well-off segment of the population is receiving the maximum benefits from the country's rapid socio-economic advancement over the past years.

Unlike the developed countries, the concept of intergenerational mobility remains a relatively less explored area of research in developing countries [27]. In Bangladesh, no significant scientific research has been conducted to assess the intergenerational mobility considering the stochastic modelling approach. With this backdrop, this research intends to evaluate the intergenerational mobility in Bangladesh across generations. A stochastic modeling approach is used to measure the intergenerational mobility, which relies on the change in educational attainment over the generations. The reason for using education in the measurement of intergenerational mobility is that 'education' is generally considered as a crucial factor for an individual's socio-economic mobility. The association between class origins and destinations of social class is largely mediated by means of educational attainment [28]. Better education can also assist socially disadvantaged populations to improve their socio-economic position [29]. Academic studies tend to focus on young people's transitions to higher education, which benefits the labor market [30] and continuing education immensely contributes to assist career development [31]. Recent researches on social mobility establish a strong association between education and intergenerational mobility [4, 32, 33].

## Materials and methods

### Measuring intergenerational mobility

Because mobility is inherently quite a complex phenomenon [1], measurement of socio-economic mobility is not straightforward and has been recognized as a challenging issue in the past. Most of the available empirical researches on intergenerational mobility usually consider three different socio-economic outcomes, e.g. income, education, and professional occupation [33]. However, irrespective of the approach used for measuring mobility, the estimates have their own methodological limitations.

Income is widely used as a measure of socio-economic status. However, income as a measure of intergenerational mobility suffers from numerous issues related to its measurements, lifetime income (permanent income) of parents and children, life cycle fluctuations, and variation of patterns of income from generation to generation. Under these circumstances, it is very difficult to find a link between their incomes. Furthermore, measuring the income of a parent causes additional bias and provides inconsistent estimates [34].

As a result, the use of nonmonetary measures e.g. education and occupations, as a proxy of the overall socio-economic status of an individual often produces useful results. But, the

occupational approach of measuring intergenerational mobility cannot be considered as bias-free due to variations in socio-economic status related to occupation over time [33]. On the contrary, education is generally considered as one of the valuable tools for improving the socio-economic wellbeing of individuals [32]. Although education is an important determinant of income, considering education in the measurement of intergenerational mobility has some limitations as well. For instance, education does not consider the quality of education and therefore can be an imperfect proxy for skill level [33]. Besides, educational attainment does not capture several other drivers affecting income level, e.g. non-cognitive abilities, parental connections to the labour market, etc. [8].

However, the use of educational attainment for research on intergenerational mobility plays an increasingly important role. Using the human capital theory, the educational attainment usually influences lifetime earnings and thus plays an important role in measuring the chances of eventual social mobility. It has been justified by the extensive literature that the higher the education level, the higher the earnings [35–37]. Thus, the evaluation of current educational inequality can also provide the basis for the evolution of future income inequality [38, 39]. With this backdrop, this research evaluates intergenerational mobility in Bangladesh by using education and investigate the change in educational attainment over the generations.

## Stochastic modeling for intergenerational mobility

In the past years, several methodologies have been proposed and adopted to evaluate intergenerational mobility, which can be broadly classified into two branches [40]. The first type of estimate utilized the income elasticity of offspring in relation to parents' income. On the other hand, the second type of measurement of social mobility utilized the concept of the Markov chain [41]. The use of the Markov chain as a tool for evaluating social mobility is old and quite common. In the Markov Chain model, mobility is measured in terms of the probability of offspring to change their socio-economic conditions as compared to their parents' one [40].

Considering the availability of credible data from a cross-sectional study, this study uses the second approach for evaluating the intergenerational mobility across Bangladeshi generations. In order to apply the Markov chain model, the study assumes that intergenerational mobility is entirely represented by the Markov chain, where the next state of the system depends only on the current state, not on any previous states [42]. This implies that the social class of the offspring depends only on the social class of their parents, not on their grandfathers' one. Moreover, the Markov chain is applicable in the case of a father having only one child. But Bangladesh not having a stable population structure is very likely that every father has an average of more than one son or daughter. Therefore, we consider only one child for each father (i.e. one respondent from each household), and use the stochastic modelling approach anticipating that our results apply in an average sense.

Suppose $\{X_n; n \geq 0\}$ is a Markov chain, where $X_n$ represents the social class of offspring at time $n$. The transition probability $P_{ij}$ of the Markov chain represents the probability of changes in social classes occurring from one generation to the next one. Let $m$ be the all possible social classes to which an individual can belong. Then, the transition probability can be defined as:

$$P_{ij} = \Pr\{X_n = j \,|\, X_{n-1} = i\}; \quad i, j = 1, 2, \ldots, m.$$

Since the system is assumed to be closed, $\sum_{j=1}^{m} P_{ij} = 1$. Moreover, it is assumed that the probabilities are constant over time. Then transition probability matrix $P$ is an $m \times m$ matrix where each component is the $P_{ij}$ ; $i,j = 1,2,\ldots,m$. The diagonal elements of $P$ (i.e. $P_{ii}$) represent the proportion of offspring who stays in the same social class of their parents and the off-diagonal

elements (i.e. $P_{ij}$) represent the proportion of offspring who movers from their parents' social class. These concepts of transition probabilities provide the basis for measuring the change of social structure across generations. In addition, the concept of a perfect mobile society is important for the measurement of intergenerational mobility. A perfectly mobile society describes the situation that should happen ideally. In a perfectly mobile society, parents' social class would not determine the prospects of the offspring. Thus, the elements in each row of the transition probability matrix of perfect mobile society would be essentially equal and are given by the limiting distribution or equilibrium distribution of the social classes of present society under investigation [43]. The equilibrium distribution of the social class, $\pi = (\pi_1, \pi_2, \ldots, \pi_m)'$ can be obtained by solving $\pi = P'\pi$ or equivalently, $\lim_{n \to \infty} P^n = \pi'$ [42]. Here, $\pi'$ is independent of the unit of time in which the elements of $P$ are measured.

## Social mobility index

This research utilizes both the individual class mobility and aggregated social mobility indices. The social mobility of each individual class is measured as the ratio of the average time spent in a social class in present society to the average time spent in a social class in a perfect mobile society [44]. Thus, the measure of the mobility of the $j^{th}$ social class, $M_j$ is given by:

$$M_j = \frac{1 - \pi_j}{1 - P_{jj}} \quad ; \quad j = 1, 2, \ldots, m.$$

For a perfect mobile society, $M_j$ takes the value 1, and the higher value of $M_j$ (i.e. $M_j > 1$) indicates lower social mobility. On the other hand, the measures of immobility for the $j^{th}$ class are then defined as $\frac{P_{jj}}{\pi_j}$ ; $j = 1, 2, \ldots, m$.

While individual class mobility indices assess the contribution of each social class in the change of social structure across generations, the aggregated social mobility evaluates the overall mobility status in the society. The aggregate mobility, in this research study, is measured by using the average probability across all social classes that an offspring will leave her/his parent's class [43, 45]:

$$M_T = 1 - \frac{(Trace\ P)}{m}.$$

The index $M_T$ lies between 0 and 1. It takes the value zero for a perfectly immobile society and unity for perfectly mobile society. In contrast, the immobility ratio is defined as $1 - M_T$.

Another mobility index considering the average class boundaries crossed over the generations [46] expressed as:

$$M_B = \frac{1}{m} \sum_{i=1}^{m} \sum_{j=1}^{m} p_{ij} |i - j|.$$

The non-negative index $M_B$ considers the distance travelled by the movers. It takes on the value 0 in the case of perfect immobility.

## Description of data

**Data sources.**   This research utilizes a nationally representative sample survey of 8,403 respondents, of which 5,436 respondents were taken from rural areas and the rest 2,967 from urban areas. Both male and female respondents aged 23 years and above were included in the sample. We also considered one child from each father keeping in mind the properties of the

transition probability matrix. The survey was conducted by Research, Training and Management (RTM) International by using a two-stage cluster sampling approach.

**Transition probability matrix from survey data.** In order to measure the intergenerational mobility from survey data, it is important to construct the transition probability matrix of the social classes. The Maximum Likelihood Estimators of the transition probabilities can be obtained from:

$$\hat{P}_{ij} = \frac{n_{ij}}{n_{i.}} \; ; \; i,j = 1,2,\ldots,m$$

where, $n_{ij}$ is the number of offspring in social class $j$ whose parents were in social class $i$ and $n_{i.} = \sum_{j=1}^{m} n_{ij}$ is the initial number of people in state $i$ and obviously, $n = \sum_{i=1}^{m} n_{i.}$ is the total number of the individuals in the sample [47].

## Results

### Background characteristics

Among the 8,403 respondents interviewed, 52.7 percent are female, and the remaining 47.3 percent are male. Overall, 36.4 percent of the respondents belong to the age group of 25–34 years with slight differences among rural are urban areas (34.6 percent in rural areas vs. 39.7 percent in urban areas). Moreover, nearly one-third (34.4%) of the respondents belong to the age group of 35–44 years, followed by 21.3 percent belongs to 45–54 years and the remaining 8.0 percent of the respondents, age are 55 years and above (Table 1).

### The evolution of educational attainment

The analysis of the educational attainment of the respondents (children) reveals that nearly one-fourth (25.5%) are illiterate, and another 28.6 percent completed primary education. Nearly, one-third (33.2%) completed secondary education level, 5.5 percent competed higher secondary level and the remaining 7.3 percent attained a higher education level (graduation and above). It is also important to note that only 4.9 percent of the respondents attained higher education level, while the same is relatively higher (9.8%) among their male counterparts. Comparatively, the percentage of illiterate respondents is higher in rural areas (30.6%) as compared to urban areas (16.1%). In contrast, the percentage of higher educated respondents is notably lower in rural areas (4.1%)) than urban areas (13.1%) (Fig 1).

Altogether, more than half (58.3%) of the respondents reported that their father is illiterate, and another 20.6 percent completed primary education. In addition, only 16.3 percent of the

**Table 1. Demographic and geographic characteristics of the respondents.**

| Indicators | Overall (n = 8,403) | Rural (n = 5,436) | Urban (n = 2,967) |
|---|---|---|---|
| **Gender** | | | |
| Male | 47.3 | 47.8 | 46.5 |
| Female | 52.7 | 52.2 | 53.5 |
| **Age (in years)** | | | |
| 25–34 | 36.4 | 34.6 | 39.7 |
| 35–44 | 34.4 | 34.8 | 33.5 |
| 45–54 | 21.3 | 22.0 | 19.9 |
| 55+ | 8.0 | 8.6 | 7.0 |

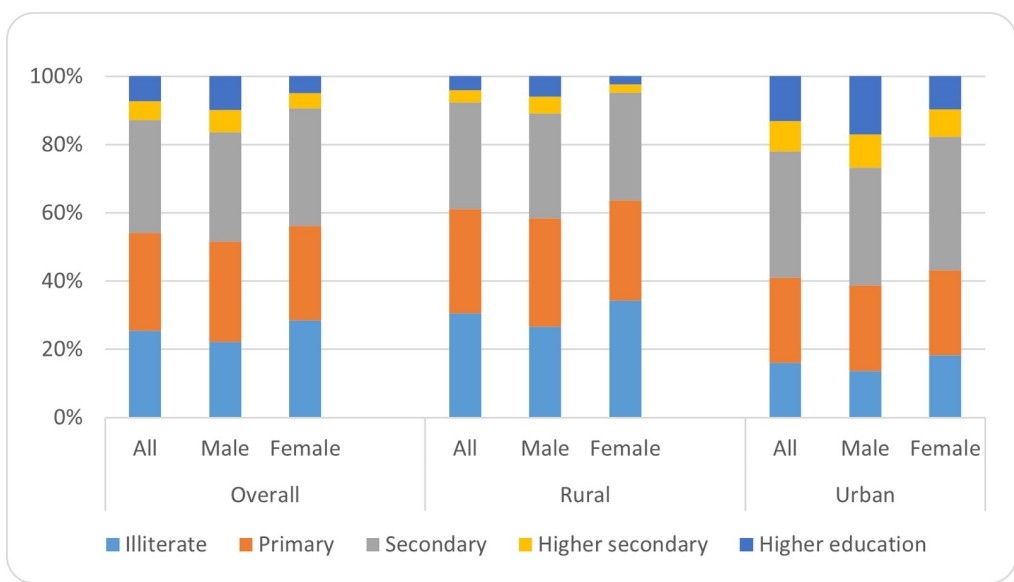

**Fig 1. Educational attainment of children.**

respondents reported that their father completed secondary education level and the rest 4.8 percent reported to complete higher secondary education or completed graduation levels. The educational attainment of the fathers also varies according to the survey locations. In rural areas, the majority (64.5%) of the respondent's fathers are illiterate, while the same is less than half (47.0%) in urban areas. The educational status of fathers in urban areas is relatively better than in rural areas (Fig 2).

When the educational attainment of children (i.e. son and daughter) is compared with the educational attainment of the father, it is revealed that 61.9 percent of the male respondents (i.e. son) and 50.1 percent of the female respondents (i.e. daughter) attains higher educational

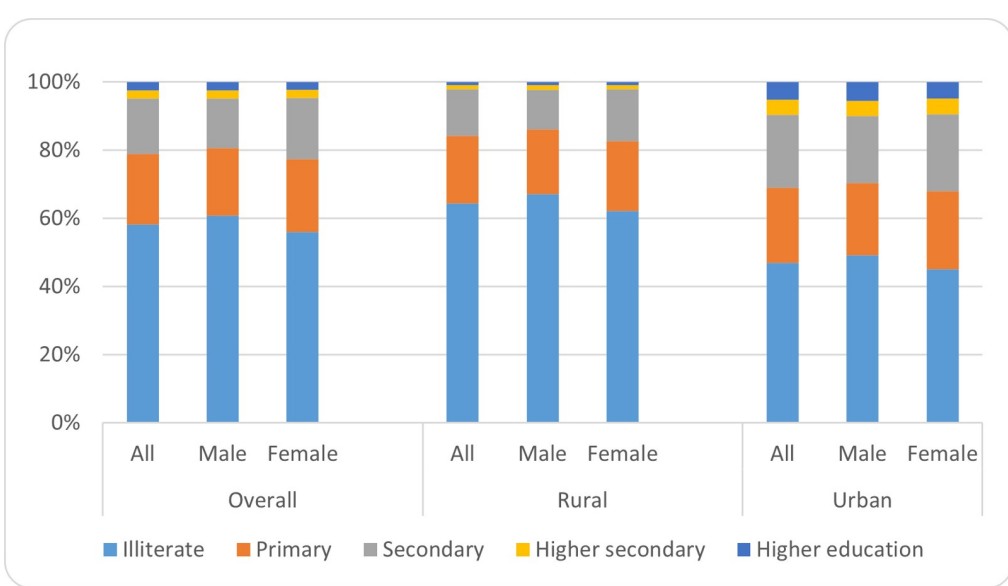

**Fig 2. Educational attainment of fathers by location and gender of the respondents.**

levels as compared to their fathers. On the other hand, 6.0 percent of the son and 11.3 percent of the daughters reported downward movement, i.e. achieves lower educational status than that of their fathers. It is noteworthy to mention that among the son 91.2 percent of the intergenerational class movement is occurred due to upward movement, while the same is 81.6 percent among the daughters. The upward movement is relatively higher among the urban children as compared to the children who are living in the rural areas. In addition, son compared to daughters achieves relatively better educational levels than their fathers (Table 2).

## Estimated transition probability matrix:

The analysis of social mobility using educational attainment of children (i.e. son and daughter) as compared to their fathers' educational status starts by setting out the transition matrix as given in Table 3. Here, the educational classes are broadly categorized as:

1. Illiterate;

2. Completed primary education;

3. Completed secondary education;

4. Completed higher secondary education; and

5. Completed graduation or post-graduation.

Thus, $i, j = 1, 2, 3, 4, 5$. The $P_{ij}$, $j^{th}$ element of $i^{th}$ row of the matrix, represents the proportion of fathers in the $i^{th}$ social class whose children move into the $j^{th}$ class in an one generation period. It is observed that the son whose fathers attain higher education also attain the same education level with a probability of 0.61. In contrast, if a father is illiterate then there is a probability of 0.37 that the children will be illiterate. More importantly, the daughters of illiterate fathers are more likely than the son to be illiterate. Also, an illiterate father will have higher educated children, with a probability of 0.02.

The educational class transition probabilities for rural and urban areas are presented in the S1 Table. It is also revealed that there exists a significant difference in educational status among the urban and rural settings. In urban areas, the probability of a higher educated father will have higher educated offspring is 0.68 which is greater than their counterpart in rural

Table 2. Intergenerational educational class movements.

| Location and gender | Stable (%) | Movement counts across different educational levels | | | |
|---|---|---|---|---|---|
| | | Upward movement (%) | Downward movement (%) | Total movement (%) | Contribution of upward movement to total movement (%) |
| Overall | | | | | |
| All | 35.6 | 55.7 | 8.8 | 64.4 | 86.4 |
| Male | 32.2 | 61.9 | 6.0 | 67.8 | 91.2 |
| Female | 38.6 | 50.1 | 11.3 | 61.4 | 81.6 |
| Rural | | | | | |
| All | 37.9 | 53.2 | 8.8 | 62.1 | 85.7 |
| Male | 34.1 | 59.8 | 6.2 | 65.9 | 90.7 |
| Female | 41.4 | 47.3 | 11.3 | 58.6 | 80.7 |
| Urban | | | | | |
| All | 31.2 | 60.1 | 8.7 | 68.8 | 87.4 |
| Male | 28.6 | 65.7 | 5.6 | 71.4 | 92.1 |
| Female | 33.5 | 55.2 | 11.3 | 66.5 | 83.0 |

**Table 3. Estimated educational class transition probabilities in Bangladesh.**

| | | State transition probabilities ($P_{ij}$) | | | | | Limiting probabilities ($\pi_j$) |
|---|---|---|---|---|---|---|---|
| | | 1 | 2 | 3 | 4 | 5 | |
| **Father to son or daughter** | 1 | 0.37 | 0.35 | 0.24 | 0.02 | 0.02 | 0.08 |
| | 2 | 0.11 | 0.24 | 0.53 | 0.06 | 0.06 | 0.15 |
| | 3 | 0.08 | 0.20 | 0.42 | 0.13 | 0.17 | 0.34 |
| | 4 | 0.01 | 0.07 | 0.36 | 0.15 | 0.41 | 0.13 |
| | 5 | 0.01 | 0.03 | 0.18 | 0.17 | 0.61 | 0.31 |
| **Father to son** | | **1** | **2** | **3** | **4** | **5** | |
| | **1** | 0.33 | 0.36 | 0.25 | 0.04 | 0.02 | 0.03 |
| | **2** | 0.05 | 0.24 | 0.54 | 0.09 | 0.08 | 0.10 |
| | **3** | 0.06 | 0.17 | 0.36 | 0.15 | 0.26 | 0.24 |
| | **4** | 0.01 | 0.08 | 0.29 | 0.16 | 0.46 | 0.14 |
| | **5** | 0.00 | 0.03 | 0.10 | 0.15 | 0.72 | 0.48 |
| **Father to Daughter** | | **1** | **2** | **3** | **4** | **5** | |
| | **1** | 0.42 | 0.32 | 0.24 | 0.01 | 0.01 | 0.12 |
| | **2** | 0.16 | 0.25 | 0.52 | 0.04 | 0.03 | 0.18 |
| | **3** | 0.09 | 0.21 | 0.47 | 0.11 | 0.12 | 0.41 |
| | **4** | 0.01 | 0.07 | 0.42 | 0.14 | 0.36 | 0.10 |
| | **5** | 0.02 | 0.02 | 0.25 | 0.19 | 0.52 | 0.18 |

areas (with a probability of 0.41). Furthermore, the probability of a higher educated father will have a higher educated child is higher in urban areas than in rural areas of Bangladesh.

The analysis of the average number of generations spent in each education class with mobility and immobility indices are presented in Table 4. The average time spent in each class (Bangladesh today) reveals that none of the education classes except higher education exceeds two generations. The longer time spent is observed for higher education with 3.54 generations for sons and 2.08 generations for daughters. Moreover, the average number of generations that would be spent in each class if the society were perfectly mobile shows that least time (1.08 generations)

**Table 4. The average number of generations spent in each social class, mobility and immobility indices in Bangladesh.**

| | Class | $P_{jj}$ | $\pi_j$ | Average no. of generations | | Mobility index | Immobility index |
|---|---|---|---|---|---|---|---|
| | | | | Bangladesh today | Perfect mobile society | | |
| **Father to son or daughter** | Illiterate | 0.37 | 0.08 | 1.60 | 1.08 | 1.5 | 4.9 |
| | Primary | 0.24 | 0.15 | 1.32 | 1.17 | 1.1 | 1.7 |
| | Secondary | 0.42 | 0.34 | 1.73 | 1.52 | 1.1 | 1.2 |
| | Higher secondary | 0.15 | 0.13 | 1.18 | 1.15 | 1.0 | 1.2 |
| | Higher education | 0.61 | 0.31 | 2.59 | 1.45 | 1.8 | 2.0 |
| **Father to son** | Illiterate | 0.33 | 0.03 | 1.49 | 1.03 | 1.4 | 10.5 |
| | Primary | 0.24 | 0.10 | 1.31 | 1.11 | 1.2 | 2.3 |
| | Secondary | 0.36 | 0.24 | 1.56 | 1.31 | 1.2 | 1.5 |
| | Higher secondary | 0.16 | 0.14 | 1.19 | 1.17 | 1.0 | 1.1 |
| | Higher education | 0.72 | 0.48 | 3.54 | 1.94 | 1.8 | 1.5 |
| **Father to daughter** | Illiterate | 0.42 | 0.12 | 1.71 | 1.14 | 1.5 | 3.4 |
| | Primary | 0.25 | 0.18 | 1.33 | 1.23 | 1.1 | 1.4 |
| | Secondary | 0.47 | 0.41 | 1.89 | 1.68 | 1.1 | 1.2 |
| | Higher secondary | 0.14 | 0.10 | 1.17 | 1.12 | 1.0 | 1.4 |
| | Higher education | 0.52 | 0.18 | 2.08 | 1.22 | 1.7 | 2.8 |

would be spent in the illiterate class. The analysis of the mobility of the individual classes also reveals that the mobility is higher in the primary, secondary, and higher secondary educational classes. In contrast, the illiterate and higher education levels are the least mobile classes.

In general, illiterate fathers are less likely to send their sons to schools as compared to their daughters with an immobility index 10.5 for sons versus 3.4 for daughters. Such difference is more evident in the urban areas where it is highly likely that sons of an illiterate father are also illiterate. This may be because illiterate parents are poor and involve their sons, for their subsistence, in income-generating activities at their early childhood (Fig 3).

The comparisons of mover count ratios for rural and urban areas reveal that the intergenerational mobility, which is measured in terms of educational attainments over the generations, is higher in rural areas as compared to the urban areas. Moreover, in rural areas, mobility is relatively higher (67.4%) among males as compared to their female counterparts (66.2%). In contrast, intergenerational mobility is slightly higher among females (65.6%) as compared to males (65.0%). But, when we consider the average social class moves, the mobility for males is higher than their female counterparts in both rural and urban areas (Table 5).

## Discussion

This research intends to evaluate the intergenerational mobility based on education attainment in Bangladesh. The rate of illiteracy of the respondents along with their fathers is higher in rural areas. In contrast, the educational attainment of both generations in urban areas is relatively better than in rural areas. The educational status also varies according to gender. Therefore, both geographical living location and gender are important for determining the intergenerational educational mobility. Several studies identified these factors as influential for determining intergenerational educational mobility [9, 48].

The findings reveal that the intergenerational upward movement is six times higher than downward movement. Moreover, the upward movement is relatively higher among males. This is because, in a country like Bangladesh, sons receive preference over daughters to receive higher-level education because of existing social values. In addition, parents want their sons to receive higher education levels as compared to their own education level because sons are

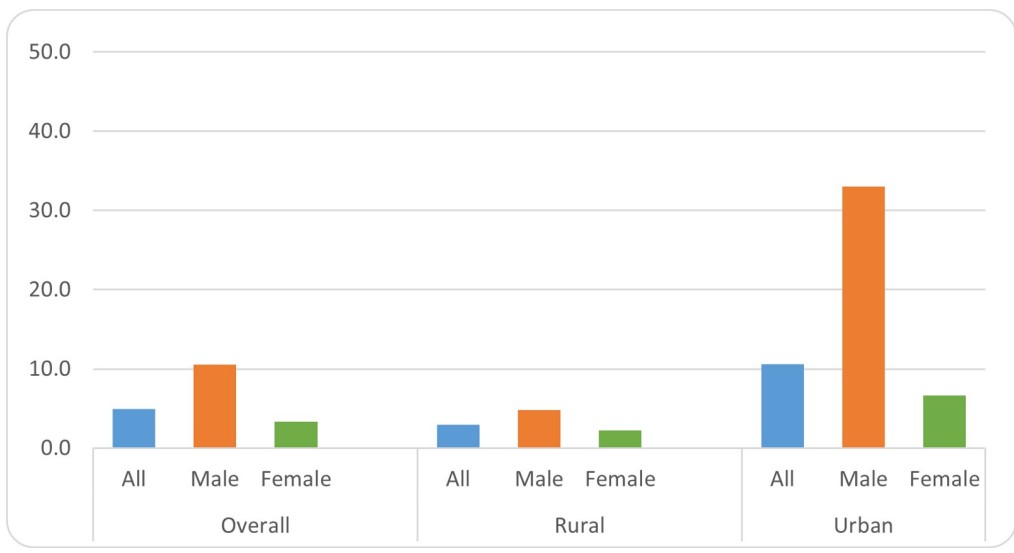

**Fig 3. Immobility indices for the illiterate class by location and gender.**

**Table 5. Estimates of Prais-Bibby and Bartholomew indices of mobility.**

| Location and gender | Immobility ratio (%) | Prais-Bibby Index (Mover Count Ratio) (%) | Bartholomew Index (Social class move) |
|---|---|---|---|
| **Overall** | | | |
| All | 36.1 | 63.9 | 0.86 |
| Male | 36.0 | 64.0 | 0.89 |
| Female | 35.9 | 64.1 | 0.85 |
| **Rural** | | | |
| All | 33.2 | 66.8 | 0.95 |
| Male | 32.6 | 67.4 | 0.97 |
| Female | 33.8 | 66.2 | 0.93 |
| **Urban** | | | |
| All | 34.8 | 65.2 | 0.88 |
| Male | 35.0 | 65.0 | 0.91 |
| Female | 34.4 | 65.6 | 0.86 |

considered to carry the name and family title of father and will assure parent's future [49]. In contrast, it is a general practice in rural households not to send their young daughters to schools for higher education as they believe females are born to specially run households [50]. Moreover, the intergenerational education upward movement is also slightly higher among urban children. The reasons behind that the economic status of a family, accessibility, and availability of educational institutes are better in urban areas compared to rural areas of Bangladesh. In rural areas, most of the households can afford to send at most two children to school and in such case sons receive preference due to their role as the future breadwinner for the family [49]. Social structure, as well as attitudes, may also play a key role to make this difference.

It is also evident that children whose fathers completed primary and higher secondary education are more likely to achieve higher education levels than their fathers. This finding is consistent with a previous study where it is reported that literate parents are more likely to send their children to school in Bangladesh [47, 48]. Moreover, in developing countries, children's educational attainment is more positively correlated with their parent's education [48, 49]. Evidence also shows that the likelihood of a higher educated father will have a higher educated child is higher because higher educated fathers are financially rich and they make enormous efforts to confirm their children's academic success [50, 51].

Although sons in Bangladesh usually receive more preference in attaining higher-level education than that of their father, illiterate fathers are less likely to send their son to schools as compared to their daughters and such difference is more evident in the urban areas. This may be because illiterate parents are poor and involve their sons, for their subsistence, in income-generating activities at their early childhood. Another reason could be that the government of Bangladesh offers several benefits including free education, cash incentives to attend school, increase job opportunities, etc. for enhancing the prevalence of educational status of girls [52, 53], which encourage the parents to send their girls to school [54]. On the other hand, for boys, more opportunities are available to become child labor in urban areas though child labor is not officially permitted in the country. Financially disadvantaged parents think that the opportunity costs of engaging their sons in unpaid or paid work would be higher than sending schools [55].

## Strengths and limitations

The strength of the study is the novelty of the work. It tries to fill up the gaps in the existing literature by measuring intergenerational educational mobility in Bangladesh using a stochastic

modelling approach. Though the use of regression models (e.g. semi-log or log-linear model) is common in measuring intergenerational mobility, applications of these models require access to the relevant explanatory variables such as both father's and mother's education and occupation, income, sex, religion, etc. Due to unavailability of the reliable explanatory variables, we could not adopt such regression models for analysis. The authors believe that a future study should combine different factors into a comparative study employing different techniques, for example, count regression model, stochastic model, etc. Further, it would be worthwhile to include mother's educational status in future studies with a substantially large sample size to ensure an adequate number of mothers in the higher education categories.

## Conclusion

The intergenerational educational movement in Bangladesh is generally upward. Moreover, the movement is relatively higher among males who lived in urban areas. But the children whose fathers are illiterate are relatively more likely to be illiterate and in contrast children of higher educated fathers have a comparatively greater probability of completing higher education. Overall, the illiterate and higher education levels are the least mobile classes. It is usual that the higher education class will be less mobile but to ensure the development in education it is essential to motivate illiterate parents to send their son to school instead to send them works. Also, for girls of poor illiterate families, there is a pressure from family as well as the community for early marriage when they approach adolescence [51], which needs to be eliminated.

The study generates important evidence that can be used to enhance the upward educational movement in the country. The authors anticipate that the findings will be helpful for the policymakers as the relationship between inequality and intergenerational mobility is vital for several aspects of the economic development of a country.

## Supporting information

**S1 Table. State transition probability matrices for rural and urban areas.**
(DOCX)

**S1 Dataset.**
(XLSX)

## Acknowledgments

The authors would like to express their gratitude to Research Training and Management (RTM) International for allowing them to access the dataset.

## Author Contributions

**Conceptualization:** Mohammed Nazmul Huq, Moyazzem Hossain.

**Data curation:** Faruq Abdulla.

**Formal analysis:** Mohammed Nazmul Huq, Faruq Abdulla.

**Methodology:** Mohammed Nazmul Huq, Faruq Abdulla, Sabina Yeasmin.

**Supervision:** Mohammed Nazmul Huq.

**Visualization:** Mohammed Nazmul Huq.

**Writing – original draft:** Mohammed Nazmul Huq, Moyazzem Hossain, Faruq Abdulla, Sabina Yeasmin.

**Writing – review & editing:** Mohammed Nazmul Huq, Moyazzem Hossain.

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
