## [Decision Letter · Decision Letter 0]

5 Jun 2021

PONE-D-21-13267

Measuring Social Mobility in Bangladesh: A Stochastic Modelling Approach

PLOS ONE

Dear Authors, 

Thank you for submitting your manuscript to PLOS ONE. After careful consideration, we feel that it has merit but does not fully meet PLOS ONE’s publication criteria as it currently stands. Therefore, we invite you to submit a revised version of the manuscript that addresses the points raised during the review process.

We look forward to receiving your revised manuscript.

Kind regards,

Professor Hafiz T.A. Khan, Ph.D, CStat (UK) 

Academic Editor

PLOS ONE

Journal Requirements:

2a) If there are ethical or legal restrictions on sharing a de-identified data set, please explain them in detail (e.g., data contain potentially sensitive information, data are owned by a third-party organization, etc.) and who has imposed them (e.g., an ethics committee). Please also provide contact information for a data access committee, ethics committee, or other institutional body to which data requests may be sent.

2b) If there are no restrictions, please upload the minimal anonymized data set necessary to replicate your study findings as either Supporting Information files or to a stable, public repository and provide us with the relevant URLs, DOIs, or accession numbers. For a list of acceptable repositories, please see http://journals.plos.org/plosone/s/data-availability#loc-recommended-repositories.

Reviewers' comments:

Reviewer's Responses to Questions

**Comments to the Author**

1. Is the manuscript technically sound, and do the data support the conclusions?

Reviewer #1: No

Reviewer #2: Partly

2. Has the statistical analysis been performed appropriately and rigorously? 

Reviewer #1: No

Reviewer #2: Yes

3. Have the authors made all data underlying the findings in their manuscript fully available?

Reviewer #1: Yes

Reviewer #2: Yes

4. Is the manuscript presented in an intelligible fashion and written in standard English?

Reviewer #1: Yes

Reviewer #2: Yes

5. Review Comments to the Author

Reviewer #1: The authors of this paper have attempted to analyze the social mobility in Bangladesh using the Markov Chain model. Scientific research related to social mobility in Bangladesh using a probabilistic approach is hardly seen in the existing literature, and filling this gap is the novelty of this work. However, I have some important comments regarding their work which are mentioned below:

(i) This study considers that the socio-economic conditions of offspring depend only on their father's socio-economic conditions. But in reality, offspring socio-economic conditions depend not only on their father's socio-economic conditions but also mother socio-economic conditions, sibling’s socio-economic conditions, gender, region and etc. Even this is evident in the descriptive statistics provided in Table 2. The authors may use multiple logistic regression as a very basic model before considering any sophisticated model which offers the flexibility to accommodate other covariates as well along with father's socio-economic conditions to model offspring socio-economic conditions.

(ii) Under the current Markov Chain framework where offspring socio-economic conditions depend only on their father's socio-economic conditions is treated as the simplest type of model to analyze social mobility (Borah. S (2013)). It would be really interesting work if authors can come up with new modelling tricks to model offspring socio-economic conditions which can take into account for other covariates as well.

Smita Borah, Stochastic Modelling of Social Mobility: A Case Study in Golaghat, Assam, International Journal of Statistics and Applications, Vol. 3 No. 3, 2013, pp. 43-49.

(iii) In Table 3, authors present a transition probability matrix for "Father to All". But the term "Father to All" is not defined anywhere before, which may make the reader confused. I guess they intended to mean all the children of a father belong to the same class at time t. For example, suppose a father has 3 children. Then all the three children will achieve the same education level at time t. If this is the case then it oversimplifies the real situation. If this is not the case then the Markov Chain does not seem applicable here.

(iv) Transition probability matrix for "Father to Son" has valid probabilistic interpretation when a father has only one son. If a father has more than one son then the Markov Chain framework fails to model this situation because of not having a valid transition probability matrix. For example, suppose in class 1 there are 15 fathers i.e. n_1=15. When one father has more than one son then ∑_(j=1)^5▒〖n_1j>15〗 which destroys the property of the transition probability matrix. Country like Bangladesh not having a stable population structure is very likely that every father has an average of more than one son. Under this condition, considering the Markov Chain to model offspring socio-economic conditions is not valid.

(v) Comment mentioned in (iv) also applicable for the transition probability matrix for "Father to Daughter"

Finally, I would like to recommend this paper for possible publication to PLOS ONE subject to the satisfactory major revision of the aforementioned points.

Reviewer #2: Upward social mobility, especially educational mobility, is an indication of improvement in a society. This nice work tries to evaluate intergenerational educational mobility in Bangladesh. The following suggestions will be helpful for improving the quality of the article.

1. Social mobility means mobility of different socio-economic factors across generations. This article deals with only one variable ‘educational status’. Here title is general but work is specific. Therefore, title should be specific like “Measuring Intergenerational Educational mobility in Bangladesh”. Again, Transition probability matrix (TPM) and Markov chain are well known stochastic approaches. So, name of the method should not mention in the title. It should be focused in methodology.

2. ‘Educational mobility’ should be highlighted in the 'introduction' especially in the objective of the study.

3. Rationale of the study should be clearly focused in the ending part of the “introduction”.

4. In Markov chain, state j and k should be matched each other. In this study, State ‘j’ represents “Father Education level” and state ‘k’ represents “offspring, son or daughter’s education level”. Here, Matching between two states is little bit confusing. Author’s reference articles are related with the intergenerational social classes (lower class, middle class, upper class etc.) which are more general and matched with each other in broader perspectives. It is better to refer work on intergenerational social mobility, especially educational mobility, between parents and offspring. Followings are some observations regarding this issue:

(i) Here transitions are: Father to all, Father to son, and father to daughter. I think limiting probability is possible to interpret Father to son only. Here, interpretation of limiting probabilities is not clear enough. It will be better to consider transition:

(a) Father to son (potential father)

(b) Mother to daughter (potential mother)

And then interpret the limiting probabilities.

(ii) For a stochastic matrix row sum must be unity. Please check the row sums in the matrices. Some miscalculations are as follows:

(a) 1st matrix (Father to all), row-5 (Table 3).

(b) 2nd matrix (Father to son), row-1 (Table 3).

(c) 3rd matrix (Father to daughter), row-1 (Table 3).

5. Some points to be noted:

(a) In table 2, column-2 label ‘Sable’ should be ‘stable’. Please correct it.

(b) “An illiterate father is three times less likely to send their sons to schools as compared to their daughters” (abstract: line no. 39-40). Without logistic regression model this finding is confusing. Please check t.

(c) Please check grammar especially comparative degree throughout the manuscript and correct accordingly.

6. PLOS authors have the option to publish the peer review history of their article (what does this mean?). If published, this will include your full peer review and any attached files.

Reviewer #1: No

Reviewer #2: No

---

## [Author Response · Author response to Decision Letter 0]

9 Jul 2021

A separate file containing the responses to reviewers comments is uploaded through editorial manager.

---

## [Editor Report · Decision Letter 1]

16 Jul 2021

Intergenerational Educational Mobility in Bangladesh

PONE-D-21-13267R1

Dear Authors,

We’re pleased to inform you that your manuscript has been judged scientifically suitable for publication and will be formally accepted for publication once it meets all outstanding technical requirements.

Kind regards,

Professor Hafiz T.A. Khan, Ph.D, CStat

Academic Editor

PLOS ONE
---

## [Editor Report · Acceptance letter]

21 Jul 2021

PONE-D-21-13267R1 

Intergenerational Educational Mobility in Bangladesh 

Dear Dr. Huq:

I'm pleased to inform you that your manuscript has been deemed suitable for publication in PLOS ONE. Congratulations! Your manuscript is now with our production department. 

Kind regards, 

on behalf of

Professor Hafiz T.A. Khan 

Academic Editor

PLOS ONE